# Treatment of Parkinson’s Disease Psychosis—A Systematic Review and Multi-Methods Approach

**DOI:** 10.3390/biomedicines12102317

**Published:** 2024-10-11

**Authors:** Olaf Rose, Sophia Huber, Eugen Trinka, Johanna Pachmayr, Stephanie Clemens

**Affiliations:** 1Institute of Pharmacy, Pharmaceutical Biology and Clinical Pharmacy, Paracelsus Medical University Salzburg, 5020 Salzburg, Austria; 2Center of Public Health and Health Services Research, Paracelsus Medical University, Strubergasse 21, 5020 Salzburg, Austria; 3Department of Neurology, Centre for Cognitive Neuroscience, EpiCARE, Christian-Doppler University Hospital, Paracelsus Medical University, 5020 Salzburg, Austria; 4Paracelsus Medical University Centre for Cognitive Neuroscience, Neuroscience Institute, Christian-Doppler University Hospital, 5020 Salzburg, Austria; 5Institute of Public Health, Medical Decision-Making and HTA, UMIT—Private University for Health Sciences, Medical Informatics and Technology, 6060 Hall in Tyrol, Austria

**Keywords:** Parkinson’s disease psychosis, pharmacotherapy, systematic review, clozapine, quetiapine, pimavanserin, survey, multi-methods

## Abstract

Objectives: Parkinson’s disease psychosis (PDP) is a prevalent non-motor symptom associated with Parkinson’s disease. The treatment options for PDP are limited, and its pharmacological management remains ambiguous. This study aimed to evaluate the existing evidence in relation to clinical practice. Methods: This multi-methods study consisted of a systematic review of reviews, adhering to the PRISMA guidelines. The review was registered with PROSPERO. Following data extraction and assessment using the AMSTAR 2 tool, a narrative synthesis was performed. In the second phase of the study, a questionnaire was developed, validated, piloted, and distributed to the heads of specialized PD clinics in Germany and Austria. Results: The search resulted in the inclusion of eleven reviews. The quality of eight of these reviews was rated as high (n = 7) or moderate (n = 1). The reviews indicated that clozapine and pimavanserin demonstrated the highest efficacy and tolerability. Other antipsychotic medications either failed to alleviate PDP symptoms or resulted in distinct motor complications. The survey findings also favored clozapine for its efficacy in managing PDP and improving quality of life, although quetiapine was regarded as effective and pimavanserin was not available. Clinicians reported initiating antipsychotic treatment at various stages of PDP, with a tendency to reduce the dosage or discontinue D2 agonists or anticholinergics. Conclusions: The reviewed literature and the survey results consistently favored clozapine for its efficacy and tolerability in treating PDP. It may be considered the first-line treatment, with pimavanserin as an alternative option.

## 1. Introduction

Parkinson’s disease (PD) is a multifaceted disorder characterized by a genetically diverse predisposition [1,2]. The primary motor symptoms are attributed to the degeneration of dopaminergic neurons within the striatum [3]. The manifestation of non-motor symptoms, which can significantly impair quality of life, is linked to corticostriatal connections and the presence of misfolded and insoluble α-synuclein in various tissue types throughout the body [4,5,6]. One prevalent non-motor complication of PD is Parkinson’s disease psychosis (PDP), which affects up to 60% of PD patients and worsens the already diminished quality of life [7]. PDP is characterized by the presence of illusions, hallucinations, and delusions, with visual hallucinations being particularly common [8]. Paranoia, delirium, and anxiety are other psychotic features, which are frequently combined, as patients with PDP usually suffer from more than one symptom [9,10]. Furthermore, these psychotic symptoms are associated with cognitive impairment, disease progression, and disability [11]. However, there is a distinct difference between dementia-related psychosis, PD-related psychosis, and schizophrenia [12]. Patients with PDP do not show the whole spectrum of schizophrenia, with all of its positive and negative symptoms. PDP symptoms are mainly limited to visual hallucinations and paranoid delusions, while, in the various dementia-related psychoses, aggression and cognitive decline are more noticeable [13]. However, PDP has a dramatic effect on the quality of life of the affected patient and is a burden to caregivers. Aamodt et al. recorded an impact on emotional health, decreased self-care, losses and disruptions in relationships, stigma, and social isolation as potential facets of stress and burden [14].

The pathophysiology of PDP is associated with the overstimulation of a decreasing number of D2-like receptors by dopaminergic drugs such as levodopa and dopamine agonists, as well as intrinsic factors related to PD itself [15]. This may elucidate the occurrence of PDP in both early and late stages of the disease, particularly when the dosages of medication are escalated. Given that most antipsychotic medications function as dopamine antagonists—contradicting the dopaminergic stimulation necessary in alleviating PD symptoms—only a limited number of second-generation antipsychotics, specifically clozapine and quetiapine, are deemed suitable for the treatment of PDP. Clozapine is a second-generation antipsychotic (SGA) with high D1, low D2, and very strong 5-HT2 receptor affinity [16]. Due to this receptor profile, the risk for tardive dyskinesia is believed to be the lowest among all antipsychotics, while the antipsychotic strength is the highest [17]. Clozapine is used in treatment-resistant schizophrenia. It is also the antipsychotic with the strongest effect on the negative symptoms of schizophrenia, such as social withdrawal or apathy. Clozapine shows considerable metabolic side effects, with weight gain, sedation, and agranulocytosis, which can potentially be lethal if not monitored. Due to likely having the most atypical receptor profile, clozapine does not affect D_2_-related motor function in PD and has been employed in PDP for many decades. Quetiapine differs from clozapine due to its more moderate metabolic effects but more pronounced sedation [18]. The risk for extrapyramidal symptoms is low, as well as for agranulocytosis. Some studies have recorded anxiolytic and antidepressant properties of quetiapine [19]. At low doses, quetiapine blocks H1 receptors, which can explain the sedation. At mid-range doses, D2 and serotonergic 5HT2A receptors are blocked. Clear antipsychotic effects are seen with high doses of 800 mg/day, when serotonergic, muscarinic, alpha-adrenergic, and histaminergic receptors are blocked. Despite some remaining D2 affinity, quetiapine has been used for many years in PDP without affecting D2-mediated motor functions [20]. Pimavanserin, a selective serotonin inverse agonist that does not interact with dopamine receptors, has been approved for use in PDP in the United States, although it lacks approval in Canada, Europe, Japan, and China [21,22]. Other second-generation antipsychotics, such as olanzapine, risperidone, and ziprasidone, are thought to retain some affinity for D2-like receptors, potentially leading to a deterioration in motor function [23].

The management of PDP remains inadequately investigated and poses significant challenges in numerous cases [24]. The reasons are the limited therapeutic options; the scarcity of evidence and guidance regarding the appropriate selection of agents for individual patients; the dilemma whereby most antipsychotics counteract motor function in PD; the high sensitivity to antipsychotic medication; the interindividual variability in the response to treatment; the heterogeneity of disease progression, etiology, and treatment (for example, patients with or without deep brain stimulation); reemergence after treatment with antipsychotics; and the overlap with PD symptoms [7,25,26]. This study addresses the clinical complexities surrounding PDP by conducting a literature review and correlating the findings with insights gathered from a survey of leading clinical specialists in PD. This multi-methods approach was chosen for multiple reasons:there are no specific guidelines on PDP;most drugs for PDP are used off-label [10];studies on medication for PDP do not reflect the heterogeneity of the aforementioned symptoms and patients;there seems to be a gap between the available drugs, literature, and clinical practice [27].

This research aims to clarify which pharmacological treatment for PDP is the most effective, safe, prevalent, and well tolerated. Additionally, it seeks to explore the rationale behind individualized treatment strategies that may justify the use of clinically controversial therapies to provide guidance in PDP treatment.

## 2. Materials and Methods

To create a higher level of validity, this multi-methods study consisted of a systematic review of reviews and a clinical questionnaire (Figure 1). The results were reflected and combined in a multi-methods matrix. The rationale behind the expansion beyond the review was to achieve adequate depth and breadth; cover clinical activities, which differ from the literature; and create more robust results via triangulation.

### 2.1. Ethics

The ethics committee of Salzburg state approved this study on 26 March 2024 (EK 1011/2024). The study was registered with the PROSPERO international prospective register of systematic reviews (CRD42024523755) and followed the revised Declaration of Helsinki [28].

### 2.2. Phase 1

A systematic review of reviews was conducted following the patient, intervention, comparison, and outcome (PICO) framework and the Preferred Reporting Items for Systematic Reviews and Meta-Analyses (PRISMA) guidelines [29]. The medical and pharmaceutical databases of PubMed, Scopus, Google Scholar, and Cochrane Library were included in the search. The search was conducted in January 2024 in the following databases.

PubMed (as of 11 January 2024): (parkinson) AND (psychosis)) AND (quetiapine)) OR (clozapine)) AND (parkinson)) AND (psychosis)) OR (pimavanserin)) AND (parkinson)) AND (psychosis).Scopus (as of 12 January 2024): (parkinson AND psychosis AND quetiapine) OR TITLE-ABS-KEY (parkinson AND psychosis AND clozapine) OR TITLE-ABS-KEY (parkinson AND psychosis AND pimavanserin.Google Scholar (as of 14 January 2024):○with all of the words “review”;○with the exact phrase “parkinson psychosis”;○with at least one of the words “quetiapin”, “clozapine”, “pimavanserin”;○where these words occur anywhere in the article.Cochrane Library (as of 14 January 2024): Parkinson AND Psychosis AND Quetiapine OR Parkinson AND Psychosis AND Clozapine OR Parkinson AND Psychosis AND Pimavanserin.

The inclusion criteria were as follows: systematic review, review, meta-analysis, publication between January 2014 and January 2024, and English or German language. The search was conducted independently by two researchers. The CITAVI^®^ Software version 6 (Swiss Academic Software GmbH, Waedenswil, Switzerland) and Endnote^®^ 21 (Microsoft, Redmond, WA, USA) were engaged for data management and for the removal of duplications. In addition, duplications were removed by hand. Studies were excluded during screening and full-text assessment if they were not written in English or German or did not match the inclusion criteria (e.g., not review). The retrieved titles and abstracts were extracted by two reviewers, who also analyzed the studies for inclusion in the review (SH, OR). Discrepancies were resolved by discussion, and the third reviewer (SC) was included in the decision-making only in a few cases. The quality of the retrieved reviews was assessed using the AMSTAR 2 tool [30,31]. Following extraction, a narrative synthesis was prepared.

### 2.3. Phase 2

To address potential remaining open questions, and as the literature on the research question seemed scarce, an expert questionnaire was developed. The design was inspired by Boynton et al. [32]. In the first step, domains were created and filled with questions. Afterwards, content validation was performed by discussing the questionnaire with four Austrian and three German specialists in the field of PD, to ensure that the questionnaire covered all essential aspects of the treatment of PDP. Finally, face validation was conducted by presenting the questionnaire to ten physicians and pharmacists, who were not specialized in PD. The questionnaire was piloted in a small group of six PD specialists, before the final version was sent out to the heads of all clinics specialized in PD in Austria (n = 32, retrieved from the Österreichische Parkinson Gesellschaft) and the 50 largest PD centers in Germany (retrieved from klinikradar.de). Informed consent was taken from all interviewees. Modifications of the questionnaire were allowed after each step. The questionnaire design featured open and closed questions, the latter with a five-point Likert-like scale or a yes/no decision. The questionnaire was transferred to SurveyMonkey and the link was sent out. Participation remained anonymous; the answers could not be tracked back to individuals. Results were imported to Microsoft Excel version 2403 and standard deviations were calculated.

## 3. Results

### 3.1. Review of Reviews

The search in the four databases led to 355 publications. After removing 79 duplications, the remaining 276 publication abstracts were assessed for inclusion. Another 205 studies did not meet the inclusion criteria. A text analysis of the remaining 71 studies yielded eleven studies, which were assessed for quality by the AMSTAR 2 tool. Another 60 studies were excluded at this step due to very low quality, not focusing on PDP or pharmacotherapy, or not being a review. The process is displayed in the PRISMA flow chart (Figure 2).

The assessment of the eleven remaining studies by the AMSTAR 2 tool showed high quality in eight reviews, while the other three reviews by Yasue et al., Abler et al., and Panchal et al. were either narrative or limited to the registration trials of pimavanserin or a pooled analysis of the pimavanserin registration trials (Table 1) [33,34,35]. The pooled analysis was funded by the manufacturer of pimavanserin [33]. These studies did not meet the criteria of the AMSTAR 2 risk of bias tool but were included in the narrative synthesis.

Heterogeneity between the studies was low, as the original studies included in the reviews were largely overlapping. This is especially true for the more recent Pimavanserin studies. Studies often used the Brief Psychiatric Rating Scale (BPRS) or the Clinical Global Impression Rating Score (CGI-S) for efficacy and the Unified Parkinson’s Disease Rating Scale part III (UPDRS-III) for adverse motor effects. Studies were excluded from the synthesis, if they were of very low quality in the AMSTAR 2 assessment or didn’t provide answers to the research question. Excluded studies and their AMSTAR 2 evaluation are listed in Appendix A. Table 2 summarizes the results of the examined interventions of the included reviews.

### 3.2. Study Results in Detail

#### 3.2.1. Clozapine, Quetiapine, and SGAs

Iketani et al. conducted a comprehensive meta-analysis and included a large variety of second-generation antipsychotics and pimavanserin [36]. The quantitative synthesis in this work was performed with 17 studies. Efficacy was tested by the BPRS, CGI-S, and UPDRS III and the reasons for dropouts were analyzed. Iketani et al. found clozapine to be effective against PDP, with a low impact on motor function, but there were safety issues regarding agranulocytosis. Quetiapine was found to be inferior to clozapine but superior to a placebo, with a good safety profile. Ziprasidone and risperidone were the most effective and risperidone was tolerated the best. Efficacy was also seen with olanzapine. However, these three drugs were believed to impair motor function and were not recommended. Rivastigmine was mentioned as a therapeutic option. Jethwa et al. found no efficacy for quetiapine in their meta-analysis [37]. Clozapine was effective; agranulocytosis was reported to happen in about 3% of patients. They also examined studies on olanzapine but found no efficacy on PDP, with a detrimental deterioration in motor scores, based on three studies, which were heterogeneous and partially did not include adverse events. Wilby et al. had a wide scope in their review and analyzed studies on quetiapine, clozapine, olanzapine, and pimavanserin [42]. For quetiapine and olanzapine, they found little efficacy for PDP, with motor side effects in some cases. Studies on clozapine lasted a maximum of 22 weeks [44]; those on quetiapine were limited to 12 weeks [45].

#### 3.2.2. Pimavanserin

Pimavanserin was less effective compared to clozapine in this meta-analysis but had a favorable safety profile. As a summary, Iketani et al. suggested the use of clozapine, due to high efficacy and low motor adverse effects. In the case of concerns about agranulocytosis, pimavanserin and quetiapine were mentioned as alternatives.

Mansuri et al. and Yasue et al. updated the review of Iketani et al. with a focus on pimavanserin and found similar results in the same four studies [35,41]. The efficacy was significant, measured by different sets of the “Scales for Assessment of Positive Symptoms” (SAPS). They concluded that further studies were needed to replicate the efficacy and safety and that no long-term data were available. A review by Kitten et al. also solely included pimavanserin studies and found it to be effective regarding improvements in the CGI-I, caregiver burden, nighttime sleep, and daytime wakefulness [39]. Abler et al. added the aspect that pimavanserin was also tolerated well in patients with PDP and dementia. The examined study period here lasted up to 9 months and it found only a few adverse drug reactions.

Jethwa et al. found pimavanserin to be effective in a review [37]. Yunusa et al. reported efficacy for clozapine and pimavanserin for PDP symptoms but not for quetiapine, which was found to impair cognition and should be avoided according to the study results [38]. A review by Chen et al. focused on randomized controlled trials on clozapine and quetiapine [40]. In contrast to clozapine, they saw little or no efficacy for quetiapine in PDP. Quetiapine was tolerated well, with confusion, dizziness, headache, orthostatic hypotension, somnolence, and worsening parkinsonism as rare side effects.

Pimavanserin was effective in a review by Wilby et al. and was claimed to be the first drug with a proven effect on the caregiver burden [42]. Similar to the studies of Heim et al. and Mansuri et al., Wilby et al. pointed out that the pimavanserin studies were conducted over a very short duration of 4–6 weeks only.

### 3.3. Questionnaire

In the period of data collection for the questionnaire conducted in February and March 2024, a total of 24 responses were obtained from PD specialists, comprising 17 respondents from Germany and seven from Austria. The mean age of the participants was 50.6 years, with the youngest participant being 39 years old and the oldest 71 years old. The gender distribution included four female participants and 19 male participants, while one individual opted not to disclose their gender.

Utilizing a Likert-like scale ranging from one to five, the specialists identified visual hallucinations as the most prevalent symptom associated with PDP, receiving a rating of 4.5 (SD = 0.8). This was followed by agitation, rated at 3.7 (SD = 0.9), delusions at 3.4 (SD = 0.9), and auditory hallucinations at 2.0 (SD = 0.9). Other symptoms and signs were not specified in this report. The perceived efficacy of quetiapine for the treatment of PDP was rated at 2.7 (SD = 0.8), while clozapine was rated more favorably at 4.0 (SD = 0.5). Adverse events were reported more frequently with quetiapine, receiving a rating of 4.5 (SD = 1.1), compared to clozapine, which was rated at 3.9 (SD = 1.3). Detailed findings regarding the efficacy and adverse events associated with these medications are illustrated in Figure 3.

The use of other, most likely erroneously prescribed antipsychotics for use in PDP was seen by all PD specialists. This happened rarely or very rarely in 52% of the raters, sometimes for 39%, and frequently for 9% of the participants. Some specialists, however, considered the use of risperidone if it was tolerated well. The prescription of other antipsychotics was mentioned by very few clinical specialists, if patients were hospitalized or agitated. Overall, 77% of clinical specialists stated that the use of other antipsychotics other than clozapine or quetiapine results in the deterioration of motor function.

In the case of PDP, most clinical specialists would primarily reduce or discontinue doses of dopamine agonists (54%) or anticholinergics (46%). In PDP patients with cognitive impairment, most specialists would still use quetiapine, despite an explicit guideline recommendation to use clozapine (79%). One specialist stated that he would also use rivastigmine in these patients. Almost all clinical specialists mentioned that they used antipsychotics beginning with mild symptoms of PDP (87%); the others used them only in severe cases. Exactly half of the experts agreed that the efficacy of quetiapine is limited; the others stated that the efficacy of quetiapine is usually good. In more detail, 38% of specialists use quetiapine only in mild cases of PDP, 33% always use it as the first choice, 13% use it if agranulocytosis monitoring is not ensured, and the remaining 17% use it in delusions and agitation, being preferable in older patients or those with cognitive impairment and for fast relief. On the other hand, 76% of them prefer clozapine in patients with severe PDP symptoms or if therapy with quetiapine is not sufficient, 14% in all cases where blood tests can be performed, and 14% in younger patients or in patients with PDP and additional strong tremors. Somnolence was the most frequently mentioned adverse effect for quetiapine and clozapine alike. The frequencies of the rated adverse events are shown in Table 3.

Almost all PD specialists stated that they would prescribe antipsychotics to all patients, regardless of the strength and nature of the symptoms (80%). The reasons not to prescribe an antipsychotic included the following: a dose reduction for dopaminergic drugs was effective, the QTc time prolongation was recorded, patients had a low leukocyte count, and new drugs were given and the dose could be decreased. The PD specialists were also asked in which patients they preferred quetiapine. The statements were “in mild psychosis”, “if leukocyte count cannot be performed”, “at the initial phase and in agitated patients”, “in milder cases”, “always my first choice”, “older patients with cognitive impairment”, “in nursing home residents”, “for practical reasons despite low evidence”, and “in patients with delusions”.

Figure 4 provides an overview on the study results for both phases.

## 4. Discussion

This review revealed eleven prior systematic reviews concerning the application of clozapine (n = 7), olanzapine (n = 5), pimavanserin (n = 9), and quetiapine (n = 7) in the treatment of PDP, with the majority demonstrating high methodological rigor. The introduction of pimavanserin as a therapeutic option for PDP in 2016 appears to have spurred much of the subsequent research activity in this area. Notably, there were no underlying direct comparative trials, with the exception of two studies that evaluated clozapine against quetiapine, as discussed in the review by Chen et al. [40]. Indirect comparisons were facilitated through the meta-analyses included. All reviews indicated that both clozapine and pimavanserin demonstrated efficacy in the treatment of PDP, without resulting in motor complications. The risk of agranulocytosis associated with clozapine was estimated to affect approximately 1–3% of the patients receiving treatment [40,46]. In contrast, olanzapine exhibited no beneficial effects on PDP, and, in certain instances, it was even associated with the worsening of motor symptoms [37]. This result aligns with the assessment provided by the clinical specialists, who indicated that all other antipsychotic medications exacerbate motor symptoms. However, this contrasts with the assertion found in the German guidelines on PD, which characterizes olanzapine as a medication devoid of motor complications. Nonetheless, it is noted that olanzapine may produce anticholinergic effects, making it a less favorable option for the treatment of PDP [47].

Risperidone and ziprasidone were effective but impaired motor function. Quetiapine was tolerated well in most reviews but the PDP symptoms did not improve significantly compared to a placebo, according to most reviews. Furthermore, one study recorded detrimental effects on cognition. The interpretation of this review of reviews hence leads to the clear conclusion that only clozapine and pimavanserin should be used to treat PDP. This result opposes common clinical practice, where quetiapine is frequently used off-label in this indication [10].

Pimavanserin, with its good efficacy, rare adverse events, and no impairment in motor function, can be an alternative for clinicians in the United States, if the risk for agranulocytosis outweighs the benefits of the higher efficacy of clozapine. Given its non-availability in the rest of the world, clozapine remains the only clearly effective treatment for PDP that does not impair motor function.

The findings from the questionnaire exhibit a notable divergence from those derived from the review of reviews. While the majority of PD specialists indicated that clozapine is more efficacious and better tolerated than quetiapine, the utilization of both medications was reported to be comparable. This phenomenon may be attributed to the significant risk of agranulocytosis associated with clozapine, which can lead to life-threatening complications [46]. Notably, it was mentioned by the clinicians that clozapine-induced agranulocytosis was a frequent experience for them and that leukocyte monitoring cannot always be warranted. This seems to be the case in nursing home residents. Some specialists stated that they preferred quetiapine not only in these patients, but whenever a patient is highly agitated or when delusions are the major symptom. The frequent use of quetiapine has been observed in the FINPARK study [27]. Interpreting the statement on preferred quetiapine use, it is found that, in some clinical settings, blood counts seem to be difficult to perform. Some PD specialists might take advantage of the calming and sedative effect of quetiapine more than the antipsychotic effects [18]. Others seem to balance the risk versus benefit of drug treatment in milder cases and favor the safer quetiapine. The inconsistent availability of leukocyte monitoring in certain clinical environments is the most significant obstacle to the utilization of clozapine. Nevertheless, in severe cases, clozapine is distinctly favored and quetiapine is preferred for use in milder cases. An intriguing observation is that some PD specialists appear to prescribe risperidone in specific scenarios, despite its negative effects on motor function [48]. In these instances, the deterioration in PD symptoms is deemed acceptable in order to alleviate the symptoms of PDP.

In summary, the results from the two phases indicate that this review of reviews identified efficacy solely for clozapine and pimavanserin, whereas clinicians continued to prescribe quetiapine despite its lower estimated efficacy and the higher incidence of adverse events compared to clozapine. The overwhelming majority of the clinical specialists expressed willingness to utilize antipsychotics at any stage of PDP. The German guideline on PD relies on the review by Yunusa et al. as the most recent source of evidence [38,47]. It recommends the use of clozapine and alludes to the controversial use of quetiapine, which results from the problem of potentially occurring agranulocytosis caused by clozapine and a lack of better alternatives.

### Limitations

This review of reviews was limited to hits during the past ten years; however, older studies were included in these publications. Even though the quality of the reviews was high, some of them mentioned potential biases in the underlying studies. The questionnaire was performed only with a small number of PD specialists and was addressed to the heads of the departments. This imposed a selection bias, which was intended to include the expertise of experienced PD specialists and reflects the high degree of specialization needed in the care of PD. However, a larger sample and a broader perspective may have led to different results. The multi-methods design supported a broad view, as the literature alone did not cover the clinical implications. The limited number of PD specialists does not allow us to generalize their statements. However, it provides practitioners with insights for clinical practice—for example, why quetiapine is used in certain cases despite lacking evidence.

## 5. Conclusions

In summary, the findings from this review of the existing literature indicate that clozapine should be considered the primary treatment option for patients with PDP at any stage, particularly when the dosages of D2 agonists or anticholinergics cannot be diminished. Pimavanserin emerges as the most viable alternative; however, its availability is currently restricted to the United States. In contrast, quetiapine demonstrated minimal to negligible efficacy, notwithstanding its common prescription. Clinical observations have corroborated clozapine’s superiority in terms of both its effectiveness and the profile of adverse events. Nevertheless, the utilization of quetiapine has been more prevalent than anticipated, with claims of its efficacy likely stemming from concerns regarding the agranulocytosis risk associated with clozapine, compounded by the inconsistent availability of leukocyte monitoring. Based on the results of this study, clozapine, which offers enhanced efficacy for PDP and improved quality of life alongside better tolerability, should be prioritized. Consequently, it is imperative that blood sampling for leukocyte monitoring be made accessible to all patients with PDP across various healthcare settings to ensure optimal care and address existing challenges. This study did not incorporate the patient perspective on pharmacotherapy for PDP, an aspect that warrants consideration and may yield additional insights. While initiatives aimed at enhancing patient care are underway, there is a pressing need for more patient-centered and longitudinal research to further inform treatment strategies.

## Figures and Tables

**Figure 1 biomedicines-12-02317-f001:**
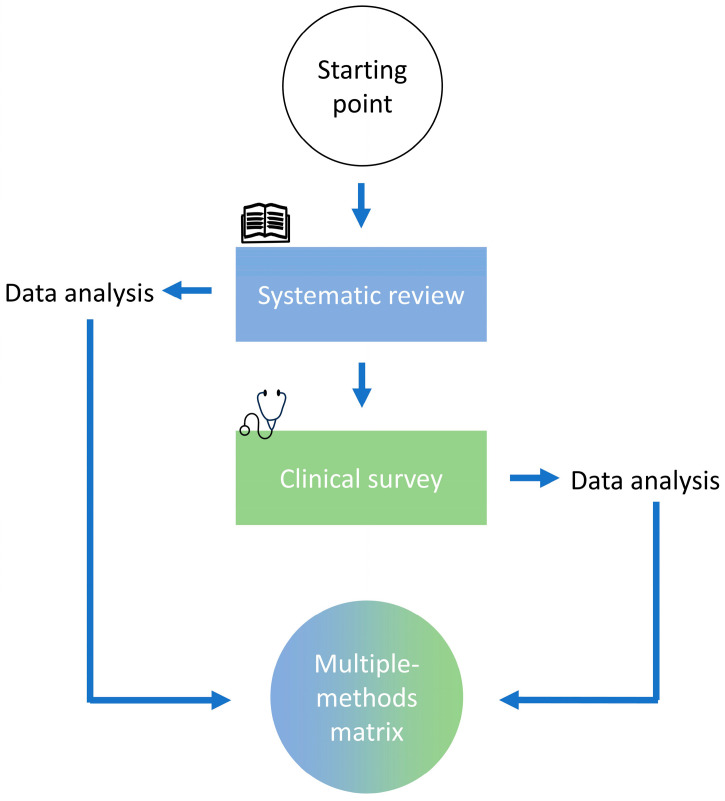
Flowchart of the multi-methods study design.

**Figure 2 biomedicines-12-02317-f002:**
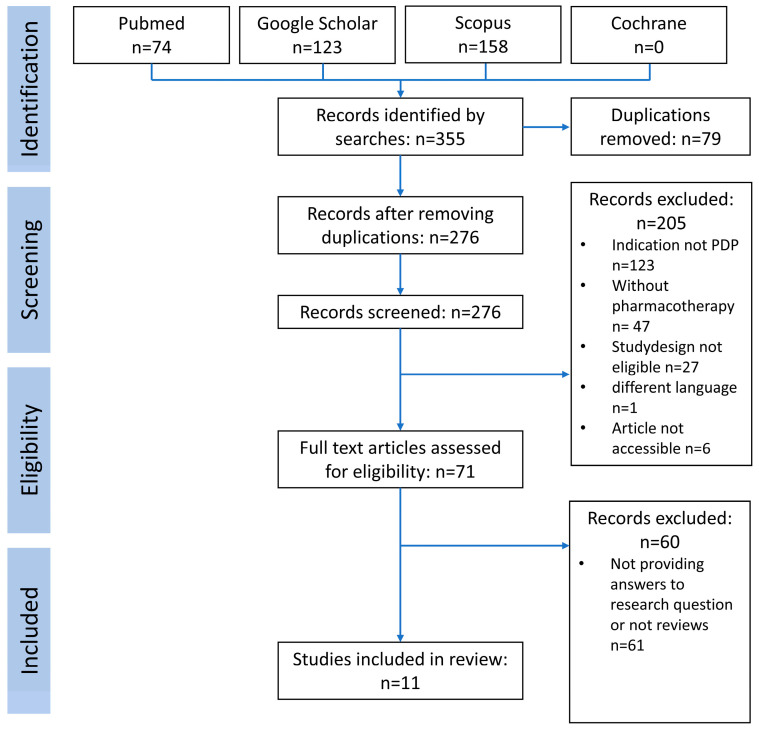
PRISMA flow chart of the systematic review.

**Figure 3 biomedicines-12-02317-f003:**
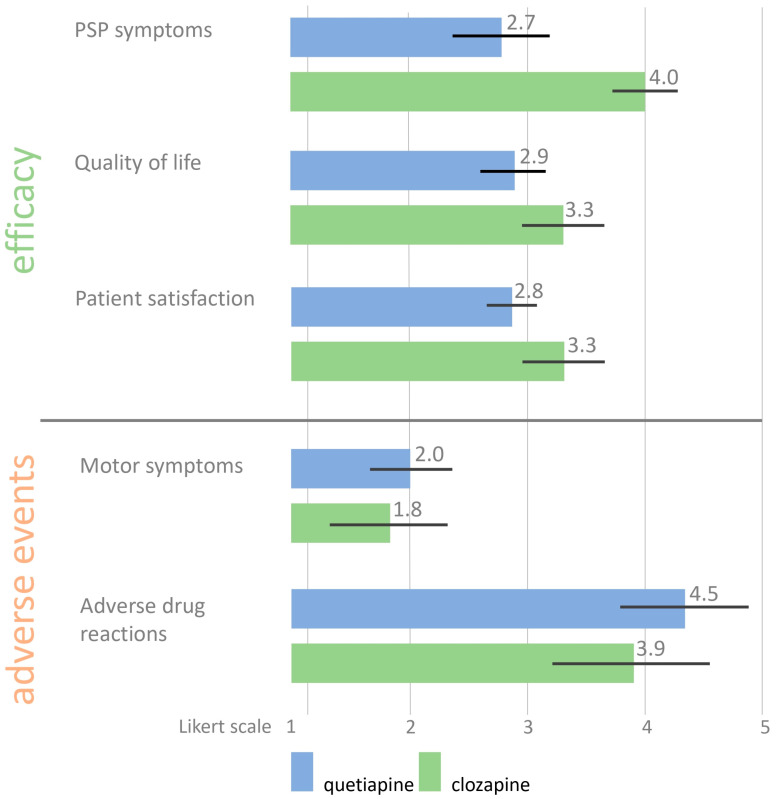
Efficacy and adverse events of clozapine and quetiapine as rated by Parkinson’s disease clinical specialists (n = 24). PDP = Parkinson’s disease psychosis.

**Figure 4 biomedicines-12-02317-f004:**
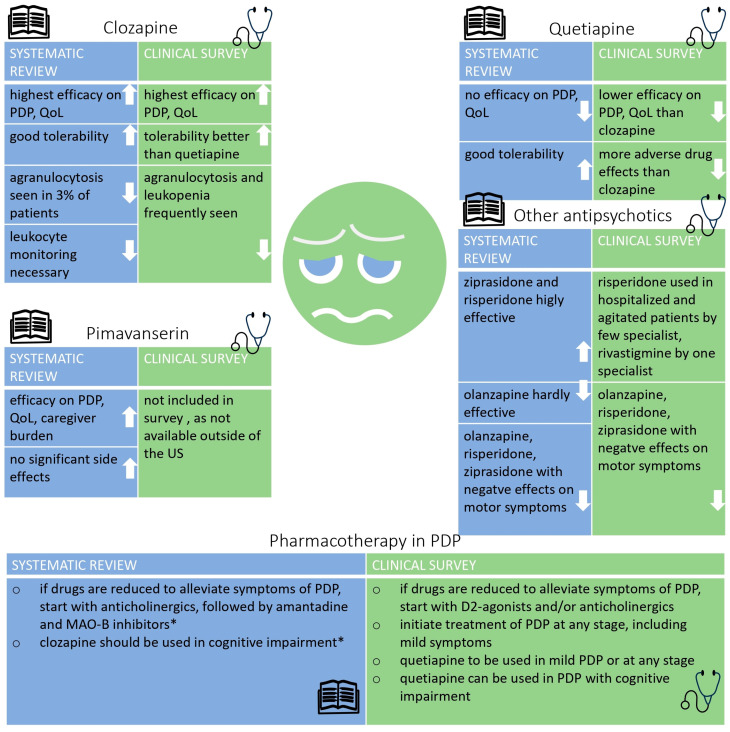
Combined results of the systematic review and the clinical survey for the drugs used in PDP. QoL = quality of life, PDP = Parkinson’s disease psychosis. * = according to the German guideline on Parkinson’s disease. ↑ arrow up: favors use of this drug in PDP, ↓ arrow down: does not support using this drug for PDP.

**Table 1 biomedicines-12-02317-t001:** Risk of bias assessment for the eight eligible reviews included according to AMSTAR 2.

AMSTAR 2 Item	Iketani 2020 [36]	Jethwa 2015 [37]	Yunusa 2023 [38]	Kitten 2018 [39]	Chen 2019 [40]	Mansuri 2022 [41]	Wilby 2017 [42]	Iketani 2017 [43]
Did the research questions and inclusion criteria for the review include the components of PICO?	No	Yes	Yes	No	No	No	No	No
Did the report of the review contain an explicit statement that the review methods were established prior to the conduct of the review and did the report justify any significant deviations from the protocol?	No	No	No	No	No	No	No	No
Did the review authors explain their selection of the study designs for inclusion in the review?	Yes	Yes	Yes	Yes	Yes	Yes	Yes	Yes
Did the review authors use a comprehensive literature search strategy?	Yes	Yes	Yes	Yes	Yes	Yes	Yes	Yes
Did the review authors perform study selection in duplicate?	Yes	Yes	Yes	Yes	Yes	N/A	Yes	Yes
Did the review authors perform data extraction in duplicate?	Yes	Yes	Yes	Yes	Yes	N/A	Yes	Yes
Did the review authors provide a list of excluded studies and justify the exclusions?	Yes	Yes	Yes	No	No	No	No	No
Did the review authors describe the included studies in adequate detail?	Yes	Yes	Yes	Yes	Yes	Yes	Yes	Yes
Did the review authors use a satisfactory technique for assessing the RoB in individual studies that were included in the review?	No	Yes	Yes	No	Yes	Yes	Yes	Yes
Did the review authors report on the sources of funding for the studies included in the review?	Yes	Yes	Yes	No	Yes	No	Yes	No
If a meta-analysis was performed did the review authors use appropriate methods for statistical combination of results?	Yes	Yes	Yes	N/A	N/A	Yes	N/A	Yes
If a meta-analysis was performed, did the review authors assess the potential impact of RoB in individual studies on the results of the meta-analysis or other evidence synthesis?	Yes	Yes	Yes	N/A	N/A	Yes	N/A	Yes
Did the review authors account for RoB in individual studies when interpreting/discussing the results of the review?	No	No	No	No	No	Yes	Yes	No
Did the review authors provide a satisfactory explanation for, and discussion of, any heterogeneity observed in the results of the review?	Yes	Yes	Yes	Yes	Yes	Yes	Yes	No
If they performed quantitative synthesis, did the review authors carry out an adequate investigation of publication bias (small study bias) and discuss its likely impact on the results of the review?	Yes	Yes	Yes	N/A	N/A	Yes	N/A	Yes
Did the review authors report any potential sources of conflicts of interest, including any funding they received for conducting the review?	Yes	Yes	Yes	No	Yes	No	Yes	Yes
AMSTAR 2 quality score	High	High	High	Moderate	High	High	High	High

N/A = not available or not applicable, RoB = risk of bias.

**Table 2 biomedicines-12-02317-t002:** The extraction of the included reviews.

Study, Year	Study Design	Number of Included Studies	PDP Drugs Covered	Outcome Measures	Results on PDP	AMSTAR 2 Quality Rating
Iketani, 2020 [36]	meta-analysis	14	clozapine, olanzapine, pimavanserin, quetiapine, risperidone, rivastigmine, ziprasidone	BPRS, CGI-S, UPDRS-III	clozapine most effective, pimavanserin effective and safe	high
Yasue, 2016 [35]	meta-analysis	4	pimavanserin	SAPS	pimavanserin effective and safe	N/A
Jethwa, 2015 [37]	meta-analysis	9	clozapine, olanzapine, pimavanserin, quetiapine	BPRS, CGI-S, UPDRS-III	clozapine and pimavanserin best options	high
Yunusa, 2023 [38]	meta-analysis	19	clozapine, pimavanserin, quetiapine	CGI-S	clozapine and pimavanserin best options	high
Kitten, 2018 [39]	review	4	pimavanserin	SAPS, CGI-S, CGI-I, caregiver burden, nighttime sleep, daytime wakefulness	pimavanserin effective in all measures	moderate
Chen, 2019 [40]	systematic review	7	clozapine, quetiapine	BPRS, UPDRS-III	quetiapine not effective	high
Mansuri, 2022 [41]	meta-analysis	4	pimavanserin	SAPS, UPDRS II and III	pimavanserin effective, also against orthostatic hypotension	high
Wilby, 2017 [42]	systematic review	16	clozapine, olanzapine, pimavanserin, quetiapine	PANSS, UPDRS III, BPRS, MMSE, Hamilton Rating Scale for Depression, Epworth Sleepiness Score, CGI	clozapine and pimavanserin best options	high
Iketani, 2017 [43]	meta-analysis	10	clozapine, olanzapine, quetiapine, risperidone	UPDRS III, BPRS	clozapine and risperidone effective, olanzapine, quetiapine and risperidone showed deterioration in motor function	high
Abler, 2022 [33]	review	3	pimavanserin	ESRS-A, UPDRS III	pimavanserin safety shown	N/A
Panchal, 2018 [34]	narrative review	several citations	aripripazole, clozapine, olanzapine, pimavanserin, quetiapine, risperidone, rivastigmine, ziprasidone		clozapine and pimavanserin best options, all other drugs showed deterioration in motor function	N/A

BPRS = Brief Psychiatric Rating Scale, CGI = Clinical Global Impression Scale, ESRS-A = Extra-Pyramidal Symptom Rating Scale, SAPS = Scale for Assessment of Positive Symptoms, UPDRS = Unified Parkinson’s Disease Rating Scale.

**Table 3 biomedicines-12-02317-t003:** Adverse events according to specialists’ evaluation for quetiapine and clozapine. Percentage of clinical specialists who mentioned this aspect.

Adverse Event	Quetiapine	Clozapine
Somnolence	66%	55%
Sedation	13%	25%
Vertigo	13%	-
Falls	17%	-
Impaired motor function	17%	-
Increasing PD symptoms	4%	-
Agranulocytosis	-	17%
Leukopenia	-	13%
Others	-	Salivation Delirium Changes in laboratory data

## Data Availability

The individual questionnaires are available from the corresponding author upon reasonable request.

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
