# Peer review of "Treatment of Parkinson’s Disease Psychosis—A Systematic Review and Multi-Methods Approach"

_biomedicines, 2024, doi:10.3390/biomedicines12102317_

Round 1

Reviewer 1 Report

Comments and Suggestions for Authors

1.      The introduction sets out some of the challenges present in managing PDP, but there is a lack of development about why these challenges are particularly significant. It is too general to simply state that "the management of PDP remains inadequately investigated".

2.      The introduction largely compromises valid knowledge about Parkinson’s disease and Parkinson’s disease psychosis, but it doesn’t introduce any new perspectives or questions.

3.      Some informative statements were stated without a citation such as (This study may elucidate …Jaban, and China).

4.      While the review mentions gap in the literature regarding the field of the study, the research question is not clearly mentioned in this section.

5.      The methods section includes a "systematic review of the reviews" and a "clinical questionnaire." While the authors clearly explain these two methodologies separately, it is inconsistent to combine the objective, standardized approach of a systematic review with the subjective nature of a questionnaire within the same research. It would be more appropriate to clarify this in the manuscript.

6.      Why do specialists still use quetiapine despite this relatively low rating? This gap between perception and usage needs to be analyzed more in depth.

7.      These findings represent rather specific studies that are part of the review, with only limited discussion of the generalizability of these findings to the greater population of patients with PDP. To what extent do the studies reviewed represent the overall population of PD patients-in particular, those with a varying degree of disease severity, comorbidities, or geographic regions?

Author Response

Dear reviewer, Thank you very much for your expertise and time. Your comments are much appreciated and have greatly helped to improve the quality of the manuscript. We have followed all of your advices and have amended point by point.

  1. The introduction sets out some of the challenges present in managing PDP, but there is a lack of development about why these challenges are particularly significant. It is too general to simply state that "the management of PDP remains inadequately investigated".

Thank you for pointing this out. We agree with this comment. Therefore, we have updated the text in the manuscript as:

“Paranoia, delirium and anxiety are other psychotic features, which are frequently combined, as patients with PDP usually suffer from more than one symptom (9, 10). Furthermore, psychotic symptoms are associated with cognitive impairment, disease progression and disability (11).”

  1. The introduction largely compromises valid knowledge about Parkinson’s disease and Parkinson’s disease psychosis, but it doesn’t introduce any new perspectives or questions.

Thank you for pointing this out. We agree with this comment. We have elaborated on the background and provided a rationale why this research is important:

“The reasons are limited therapeutic options, a scarcity of evidence and guidance regarding the appropriate selection of agents for individual patients, the dilemma that most antipsychotics counteract motor function in PD, a high sensitivity to antipsychotic medication, interindividual variability in response to treatment, heterogeneity of disease progression etiology and treatment (for example patients with or without deep brain stimulation), reemergence after treatment with antipsychotics and overlapping with PD symptoms (7, 17, 18). This study addresses the clinical complexities surrounding PDP by conducting a literature review and correlating the findings with insights gathered from a survey of leading clinical specialists in PD. This multi-methods approach was chosen for multiple reasons:

  1. there are no specific guidelines on PDP
  2. most drugs against PDP are used off-label (10)
  3. studies on the medication for PDP do not represent the heterogeneity of before-mentioned symptoms and patients
  4. there seemed to be a gap between available drugs, literature and clinical practice (19)”

  1. Some informative statements were stated without a citation such as (This study may elucidate …Jaban, and China).

Thank you for pointing this out. We have added more citations throughout the introduction as well as citations, which specifically refer to the FDA registration and to the situation that it is not registered anywhere else:

Mathis MV, Muoio BM, Andreason P, et al. The US Food and Drug Administration's Perspective on the New Antipsychotic Pimavanserin. J Clin Psychiatry. 2017;78(6):e668-e673.

Pagan FL, Schulz PE, Torres-Yaghi Y, Pontone GM. On the Optimal Diagnosis and the Evolving Role of Pimavanserin in Parkinson's Disease Psychosis. CNS Drugs. 2024;38(5):333-347.

  1. While the review mentions gap in the literature regarding the field of the study, the research question is not clearly mentioned in this section.

Thank you for pointing this out. We have added a clearly stated research question now:

“This research aims to clarify which pharmacological treatment for PDP is the most effective, safe, prevalent, and well-tolerated. Additionally, it seeks to explore the rationale behind clinically controversial therapies to provide guidance in PDP treatment.”

  1. The methods section includes a "systematic review of the reviews" and a "clinical questionnaire." While the authors clearly explain these two methodologies separately, it is inconsistent to combine the objective, standardized approach of a systematic review with the subjective nature of a questionnaire within the same research. It would be more appropriate to clarify this in the manuscript.

Thank you for pointing this out. This is a very crucial aspect, which we should have explained in more detail. This multi-methods approach was chosen, as the research question cannot be solved by the review alone. We have justified this in detail at bullet 2. Furthermore, we have added in the methods section:

“Results were reflected and combined in a multi methods matrix. The rationale behind the expansion beyond the review was to expand to adequate depth and breadth, cover clinical activities, which differ from literature and create more robust results by triangulation.”

  1. Why do specialists still use quetiapine despite this relatively low rating? This gap between perception and usage needs to be analyzed more in depth.

Thank you, this is a crucial point indeed. We have added the verbatim results of the PD specialists free text now to elucidate.

“PD specialists were also asked in which patient they preferred quetiapine. Statements were: “in mild psychosis”, “if leukocyte count cannot be performed”, “at the initial phase and in agitated patients”, “in milder cases”, “always my first choice”, “older patients with cognitive impairment”, “in nursing home residents”, “for practical reasons-despite low evidence”, “in patients with delusions”.

And to the discussions:

Notably it was mentioned by the clinicians that clozapine induced agranulocytosis was a frequent experience for them and that leukocyte monitoring cannot always be warranted. This seems to be the case in nursing-home residents. Some specialists stated that they preferred quetiapine not only in these patients but whenever a patient is highly agitated or when delusions are the major symptom. Frequent use of quetiapine has been observed in the FINPARK study before (19). Interpreting the statement on preferred quetiapine use indicates that in some clinical settings, blood counts seem to be difficult to perform. Some PD specialists might take advantage of the calming and sedative effect of quetiapine more than of the antipsychotic effects (38). Others seem to balance risk versus benefit of drug treatment in milder cases and favor the safer quetiapine.

  1. These findings represent rather specific studies that are part of the review, with only limited discussion of the generalizability of these findings to the greater population of patients with PDP. To what extent do the studies reviewed represent the overall population of PD patients-in particular, those with a varying degree of disease severity, comorbidities, or geographic regions?

Thank you, this should have been discussed in more detail indeed. We have added to the limitations:

“The multi-methods design supported a broad view, as literature alone didn`t cover the clinical implications. The limited number of PD specialists does not allow to generalize their statements. However, it provides practitioners with hints for clinical practice, for example why quetiapine is used in certain cases despite lacking evidence.”

Reviewer 2 Report

Comments and Suggestions for Authors

Treatment of Parkinson's disease psychosis - a systematic review and multi-methods approach

The study is interesting and the approach is innovative. You can find my appraisal and commentaries as follows:

Introduction: In the introduction, you described the principal treatments for psychotic symptoms in PD. However, I suggest adding more information about the specificity of each drug in psychiatry (e.g., clozapine) and its use to treat specific symptoms (negative ones). Moreover, you should underly the difference between dementia-related psychosis, PD-related psychosis, and that observed in psychiatric patients. Similarly, the impact of psychotic symptoms in PD patients, on their quality of life and that of caregivers, etc. is not clear. I suggest adding this. Indeed, this is a study that could be read by different professionals who could be not aware of pharmacological treatments for psychotic symptoms in PD.

Moreover, you introduced the used approach at the end of the section. However, it is unclear why this approach is useful and better than a simple systematic review. This needs to be stated in a better way.

Methods: The methods are well-explained, but I suggest adding a figure with a flowchart with the 2 phases of the study. Systematic review: the authors resolved discrepancies with a discussion. However, usually, the search of the relevant hits is performed by two authors, and then a third opinion could be required in case of discrepancies. Could you please add more info about it? This is only a little recommendation, but the method for SR was carried out in a rigorous manner. In the results, I suggest summarizing the results based on the drugs, avoiding the listing effect, without replicating the information given in the table. Moreover, I suggest to avoid to include hits that did not generate results (i.e. narrative reviews).

Questionnaire: Please, substitute Likert scale with the Likert-like scale. Likert scale is usually : (1) Strongly Disagree; (2) Disagree; (3) Neither Agree nor Disagree; (4) Agree; (5) Strongly Agree. I suppose that in your case, the scale was from 1= min to 5 max. However, it is not clear. Please, confirm. I like Fig.3, but the quality of the figure and table is satisfactory.

The discussion is interesting, but the paragraph between lines 278-288 needs references. 

Author Response

Reviewer 2

Treatment of Parkinson's disease psychosis - a systematic review and multi-methods approach

The study is interesting and the approach is innovative. You can find my appraisal and commentaries as follows:

Introduction: In the introduction, you described the principal treatments for psychotic symptoms in PD. However, I suggest adding more information about the specificity of each drug in psychiatry (e.g., clozapine) and its use to treat specific symptoms (negative ones).

Dear reviewer, Thank you very much for your expertise and time. Your comments are much appreciated and have greatly helped to improve the quality of the manuscript. We have followed all of your advices and have amended point by point.

We have added a description of both drugs to the introduction:

Clozapine is a second-generation antipsychotic (SGA) with high D1, low D2 and very strong 5-HT2 receptor affinity (16). Due to this receptor profile, the risk for tardive dyskinesia is believed to be lowest of all antipsychotics, while the antipsychotic strength is highest (17). Clozapine is used in treatment resistant schizophrenia. It is also the anti-psychotic with the strongest effect on negative symptoms of schizophrenia, such as social withdrawal or apathy. Clozapine shows considerable metabolic side effects with weight gain, sedation and agranulocytosis, which can potentially be lethal if not monitored for. Due to its probably most atypical receptor profile, Clozapine doesn`t affect D2-related motor function in PD and has been employed in PDP for many decades. Quetiapine differs from clozapine by more moderate metabolic effects but more pronounced sedation (18). Risk for extrapyramidal symptoms is low as well as for agranulocytosis. Some studies have recorded anxiolytic and antidepressant properties of quetiapine (19). At low doses quetiapine blocks H1-receptors, which can explain sedation. At mid-range doses D2 and serotonergic 5HT2A receptors are blocked. Clear antipsychotic effects are seen with high doses of 800 mg/day, when serotonergic, muscarinic, alpha adrenergic, and histaminergic receptors are blocked. Despite some remaining D2 affinity, quetiapine is used for many years in PDP without affecting D2-mediated motor functions (20). Pimavanserin, a selective serotonin inverse agonist that does not interact with dopamine receptors, has been approved for use in PDP in the United States, although it lacks approval in Canada, Eu-rope, Japan, and China (21, 22). Other second-generation antipsychotics, such as olanzapine, risperidone, and ziprasidone, are thought to retain some affinity for D2-like receptors, potentially leading to deterioration in motor function (23).

Moreover, you should underly the difference between dementia-related psychosis, PD-related psychosis, and that observed in psychiatric patients.

Thank you for this advice. We have added:

Yet, there is a distinct difference between dementia-related psychosis, PD-related psychosis and schizophrenia (12). Patients with PDP do not show the whole spectrum of schizophrenia with all its positive and negative symptoms. PDP symptoms are mainly limited to visual hallucinations and paranoid delusions, while in the various dementia-related psychosis aggression and cognitive decline are more noticeable (13).

Similarly, the impact of psychotic symptoms in PD patients, on their quality of life and that of caregivers, etc. is not clear. I suggest adding this. Indeed, this is a study that could be read by different professionals who could be not aware of pharmacological treatments for psychotic symptoms in PD.

Thank you for this advice. We have added:

However, PDP has a dramatic effect on quality of life of the affected patients and is a burden to caregivers. Aamodt et al. recorded impact on emotional health, decreased self-care, losses and disruptions in relationships, stigma and social isolation as potential facets of stress and burden (14).

And with regard to reviewer 1:

Paranoia, delirium and anxiety are other psychotic features, which are frequently combined, as patients with PDP usually suffer from more than one symptom (9, 10). Further-more, psychotic symptoms are associated with cognitive impairment, disease progression and disability (11)

Moreover, you introduced the used approach at the end of the section. However, it is unclear why this approach is useful and better than a simple systematic review. This needs to be stated in a better way.

Thank you for pointing this out. We agree with you and reviewer 1. We have added:

“The reasons are limited therapeutic options, a scarcity of evidence and guidance regarding the appropriate selection of agents for individual patients, the dilemma that most antipsychotics counteract motor function in PD, a high sensitivity to antipsychotic medication, interindividual variability in response to treatment, heterogeneity of disease progression etiology and treatment (for example patients with or without deep brain stimulation), reemergence after treatment with antipsychotics and overlapping with PD symptoms (7, 17, 18). This study addresses the clinical complexities surrounding PDP by conducting a literature review and correlating the findings with insights gathered from a survey of leading clinical specialists in PD. This multi-methods approach was chosen for multiple reasons:

  1. there are no specific guidelines on PDP
  2. most drugs against PDP are used off-label (10)
  3. studies on the medication for PDP do not represent the heterogeneity of before-mentioned symptoms and patients
  4. there seemed to be a gap between available drugs, literature and clinical practice (19)”

Methods: The methods are well-explained, but I suggest adding a figure with a flowchart with the 2 phases of the study.

Thank you for this advice. We have added a study flowchart:

Systematic review: the authors resolved discrepancies with a discussion. However, usually, the search of the relevant hits is performed by two authors, and then a third opinion could be required in case of discrepancies. Could you please add more info about it? This is only a little recommendation, but the method for SR was carried out in a rigorous manner.

Thank you for this advice. All discrepancies could be solved between the two reviewers but the third reviewer (Stephanie Clemens) was asked for advice in very few cases. We have added:

“, the third reviewer (SC) was included into decision-making only in few cases”

In the results, I suggest summarizing the results based on the drugs, avoiding the listing effect, without replicating the information given in the table.

Thank you for this advice. Structuring by drugs was our first attempt. This would work very well with single studies, but doesn`t work with the covered reviews. Otherwise, we had to mention the same review details over and over again, leading to even more repetition. However, we have amended and listed the studies under the two headings of 1. Clozapine, quetiapine and SGAs and 2. Pimavanserin, to increase readability.

Moreover, I suggest to avoid to include hits that did not generate results (i.e. narrative reviews).

Thank you, this is very true, we have deleted these studies.

Questionnaire: Please, substitute Likert scale with the Likert-like scale. Likert scale is usually : (1) Strongly Disagree; (2) Disagree; (3) Neither Agree nor Disagree; (4) Agree; (5) Strongly Agree. I suppose that in your case, the scale was from 1= min to 5 max. However, it is not clear. Please, confirm.

Thank you, we confirm and have changed to the wording Likert-like scale throughout the text.

I like Fig.3, but the quality of the figure and table is satisfactory.

We agree and our first draft was without this figure. However, many of our team members, who are not as deeply involved into PD requested a fast and understandable graphic synopsis. When we did an ‘internal peer review’, all readers recommended to keep it. This figure depicts the study results at a glance. Thank you for your understanding. 

The discussion is interesting, but the paragraph between lines 278-288 needs references. 

Thank you for pointing this out. We have changed this paragraph due to the recommendation of reviewer 1 and have added more citations.

Reviewer 2

Treatment of Parkinson's disease psychosis - a systematic review and multi-methods approach

The study is interesting and the approach is innovative. You can find my appraisal and commentaries as follows:

Introduction: In the introduction, you described the principal treatments for psychotic symptoms in PD. However, I suggest adding more information about the specificity of each drug in psychiatry (e.g., clozapine) and its use to treat specific symptoms (negative ones).

Dear reviewer, Thank you very much for your expertise and time. Your comments are much appreciated and have greatly helped to improve the quality of the manuscript. We have followed all of your advices and have amended point by point.

We have added a description of both drugs to the introduction:

Clozapine is a second-generation antipsychotic (SGA) with high D1, low D2 and very strong 5-HT2 receptor affinity (16). Due to this receptor profile, the risk for tardive dyskinesia is believed to be lowest of all antipsychotics, while the antipsychotic strength is highest (17). Clozapine is used in treatment resistant schizophrenia. It is also the anti-psychotic with the strongest effect on negative symptoms of schizophrenia, such as social withdrawal or apathy. Clozapine shows considerable metabolic side effects with weight gain, sedation and agranulocytosis, which can potentially be lethal if not monitored for. Due to its probably most atypical receptor profile, Clozapine doesn`t affect D2-related motor function in PD and has been employed in PDP for many decades. Quetiapine differs from clozapine by more moderate metabolic effects but more pronounced sedation (18). Risk for extrapyramidal symptoms is low as well as for agranulocytosis. Some studies have recorded anxiolytic and antidepressant properties of quetiapine (19). At low doses quetiapine blocks H1-receptors, which can explain sedation. At mid-range doses D2 and serotonergic 5HT2A receptors are blocked. Clear antipsychotic effects are seen with high doses of 800 mg/day, when serotonergic, muscarinic, alpha adrenergic, and histaminergic receptors are blocked. Despite some remaining D2 affinity, quetiapine is used for many years in PDP without affecting D2-mediated motor functions (20). Pimavanserin, a selective serotonin inverse agonist that does not interact with dopamine receptors, has been approved for use in PDP in the United States, although it lacks approval in Canada, Eu-rope, Japan, and China (21, 22). Other second-generation antipsychotics, such as olanzapine, risperidone, and ziprasidone, are thought to retain some affinity for D2-like receptors, potentially leading to deterioration in motor function (23).

Moreover, you should underly the difference between dementia-related psychosis, PD-related psychosis, and that observed in psychiatric patients.

Thank you for this advice. We have added:

Yet, there is a distinct difference between dementia-related psychosis, PD-related psychosis and schizophrenia (12). Patients with PDP do not show the whole spectrum of schizophrenia with all its positive and negative symptoms. PDP symptoms are mainly limited to visual hallucinations and paranoid delusions, while in the various dementia-related psychosis aggression and cognitive decline are more noticeable (13).

Similarly, the impact of psychotic symptoms in PD patients, on their quality of life and that of caregivers, etc. is not clear. I suggest adding this. Indeed, this is a study that could be read by different professionals who could be not aware of pharmacological treatments for psychotic symptoms in PD.

Thank you for this advice. We have added:

However, PDP has a dramatic effect on quality of life of the affected patients and is a burden to caregivers. Aamodt et al. recorded impact on emotional health, decreased self-care, losses and disruptions in relationships, stigma and social isolation as potential facets of stress and burden (14).

And with regard to reviewer 1:

Paranoia, delirium and anxiety are other psychotic features, which are frequently combined, as patients with PDP usually suffer from more than one symptom (9, 10). Further-more, psychotic symptoms are associated with cognitive impairment, disease progression and disability (11)

Moreover, you introduced the used approach at the end of the section. However, it is unclear why this approach is useful and better than a simple systematic review. This needs to be stated in a better way.

Thank you for pointing this out. We agree with you and reviewer 1. We have added:

“The reasons are limited therapeutic options, a scarcity of evidence and guidance regarding the appropriate selection of agents for individual patients, the dilemma that most antipsychotics counteract motor function in PD, a high sensitivity to antipsychotic medication, interindividual variability in response to treatment, heterogeneity of disease progression etiology and treatment (for example patients with or without deep brain stimulation), reemergence after treatment with antipsychotics and overlapping with PD symptoms (7, 17, 18). This study addresses the clinical complexities surrounding PDP by conducting a literature review and correlating the findings with insights gathered from a survey of leading clinical specialists in PD. This multi-methods approach was chosen for multiple reasons:

  1. there are no specific guidelines on PDP
  2. most drugs against PDP are used off-label (10)
  3. studies on the medication for PDP do not represent the heterogeneity of before-mentioned symptoms and patients
  4. there seemed to be a gap between available drugs, literature and clinical practice (19)”

Methods: The methods are well-explained, but I suggest adding a figure with a flowchart with the 2 phases of the study.

Thank you for this advice. We have added a study flowchart:

Systematic review: the authors resolved discrepancies with a discussion. However, usually, the search of the relevant hits is performed by two authors, and then a third opinion could be required in case of discrepancies. Could you please add more info about it? This is only a little recommendation, but the method for SR was carried out in a rigorous manner.

Thank you for this advice. All discrepancies could be solved between the two reviewers but the third reviewer (Stephanie Clemens) was asked for advice in very few cases. We have added:

“, the third reviewer (SC) was included into decision-making only in few cases”

In the results, I suggest summarizing the results based on the drugs, avoiding the listing effect, without replicating the information given in the table.

Thank you for this advice. Structuring by drugs was our first attempt. This would work very well with single studies, but doesn`t work with the covered reviews. Otherwise, we had to mention the same review details over and over again, leading to even more repetition. However, we have amended and listed the studies under the two headings of 1. Clozapine, quetiapine and SGAs and 2. Pimavanserin, to increase readability.

Moreover, I suggest to avoid to include hits that did not generate results (i.e. narrative reviews).

Thank you, this is very true, we have deleted these studies.

Questionnaire: Please, substitute Likert scale with the Likert-like scale. Likert scale is usually : (1) Strongly Disagree; (2) Disagree; (3) Neither Agree nor Disagree; (4) Agree; (5) Strongly Agree. I suppose that in your case, the scale was from 1= min to 5 max. However, it is not clear. Please, confirm.

Thank you, we confirm and have changed to the wording Likert-like scale throughout the text.

I like Fig.3, but the quality of the figure and table is satisfactory.

We agree and our first draft was without this figure. However, many of our team members, who are not as deeply involved into PD requested a fast and understandable graphic synopsis. When we did an ‘internal peer review’, all readers recommended to keep it. This figure depicts the study results at a glance. Thank you for your understanding. 

The discussion is interesting, but the paragraph between lines 278-288 needs references. 

Thank you for pointing this out. We have changed this paragraph due to the recommendation of reviewer 1 and have added more citations.

Round 2

Reviewer 2 Report

Comments and Suggestions for Authors

The authors addressed all my concerns.